# The Efficacy of Stem Cells in Wound Healing: A Systematic Review

**DOI:** 10.3390/ijms25053006

**Published:** 2024-03-05

**Authors:** Banu Farabi, Katie Roster, Rahim Hirani, Katharine Tepper, Mehmet Fatih Atak, Bijan Safai

**Affiliations:** 1Department of Dermatology, New York Medical College, Valhalla, NY 10595, USA; seherbf@nychhc.org; 2Department of Dermatology, NYC H+Health Hospitals/Metropolitan Hospital Center, New York, NY 10029, USA; 3Department of Dermatology, NYC H+Health Hospitals/South Brooklyn Health, Brooklyn, NY 11235, USA; 4School of Medicine, New York Medical College, Valhalla, NY 10595, USA; katieroster@gmail.com (K.R.); rhirani2@student.nymc.edu (R.H.); 5Phillip Capozzi, M.D. Library, New York Medical College, Valhalla, NY 10595, USA; ktepper@nycm.edu; 6Department of Internal Medicine, NYC H+Health Hospitals/Metropolitan Hospital Center, New York, NY 10029, USA; atakm@nychhc.org

**Keywords:** stem cells, chronic wound, wound healing, dermatology

## Abstract

Wound healing is an intricate process involving coordinated interactions among inflammatory cells, skin fibroblasts, keratinocytes, and endothelial cells. Successful tissue repair hinges on controlled inflammation, angiogenesis, and remodeling facilitated by the exchange of cytokines and growth factors. Comorbid conditions can disrupt this process, leading to significant morbidity and mortality. Stem cell therapy has emerged as a promising strategy for enhancing wound healing, utilizing cells from diverse sources such as endothelial progenitor cells, bone marrow, adipose tissue, dermal, and inducible pluripotent stem cells. In this systematic review, we comprehensively investigated stem cell therapies in chronic wounds, summarizing the clinical, translational, and primary literature. A systematic search across PubMed, Embase, Web of Science, Google Scholar, and Cochrane Library yielded 22,454 articles, reduced to 44 studies after rigorous screening. Notably, adipose tissue-derived mesenchymal stem cells (AD-MSCs) emerged as an optimal choice due to their abundant supply, easy isolation, ex vivo proliferative capacities, and pro-angiogenic factor secretion. AD-MSCs have shown efficacy in various conditions, including peripheral arterial disease, diabetic wounds, hypertensive ulcers, bullous diabeticorum, venous ulcers, and post-Mohs micrographic surgery wounds. Delivery methods varied, encompassing topical application, scaffold incorporation, combination with plasma-rich proteins, and atelocollagen administration. Integration with local wound care practices resulted in reduced pain, shorter healing times, and improved cosmesis. Stem cell transplantation represents a potential therapeutic avenue, as transplanted stem cells not only differentiate into diverse skin cell types but also release essential cytokines and growth factors, fostering increased angiogenesis. This approach holds promise for intractable wounds, particularly chronic lower-leg wounds, and as a post-Mohs micrographic surgery intervention for healing defects through secondary intention. The potential reduction in healthcare costs and enhancement of patient quality of life further underscore the attractiveness of stem cell applications in wound care. This systematic review explores the clinical utilization of stem cells and stem cell products, providing valuable insights into their role as ancillary methods in treating chronic wounds.

## 1. Introduction

Stem cells are used to modulate the progression of neurodegenerative diseases, ischemic damage in the tissues (peripheral artery disease, chronic diabetic wounds, venous ulcers, etc.), as well as for skin rejuvenation [1,2,3]. Recent studies have also shown that the secreted factors from stem cells (or cell-free extracts from stem cells) enhance various disorders through paracrine function in the tissues [4,5,6].

In the last decade, a significant rise in interest into chronic wound healing has occurred due to the increasing incidence of chronic ulcers, an aging population, the increasing incidence of diabetes, and healthcare inequalities. Non-healing ulcers can significantly reduce a patient’s quality of life and increase healthcare costs due to recurrent admissions caused by chronically infected wounds, increased risk of osteomyelitis, prolonged intravenous antibiotic treatments, amputation, and eventually the need for supportive care measures for these individuals [7,8].

In contemporary healthcare, the alarming statistic that 70% of amputations stem from unhealed wounds underscores the urgency to address this pervasive issue. With an estimated six million individuals in the United States alone grappling with the consequences of non-healing wounds, the associated healthcare expenditures have surged to a staggering USD 25 billion. In the light of these compelling figures, there is a critical need to explore innovative research avenues to mitigate the human suffering and economic burden imposed by chronic wounds. It is within this context that the exploration of stem cell therapy emerges as a promising and potentially transformative approach. Regarding research into the unique regenerative capabilities of stem cells, further research in this domain has the potential to revolutionize wound care and contribute substantially to ameliorating the impact of non-healing wounds on both individuals and the healthcare system [8,9,10,11].

Stem cells harvested from different sources can be used for wound repair and regeneration such as endothelial progenitor cells (EPCs), adult stem cells in the forms of bone marrow-derived mesenchymal stem cells (BM-MSCs), adipose tissue stem cells (ASCs), dermal stem cells (DSCs), and inducible pluripotent stem cells (iPSs). These stem cells enhance wound healing via tissue regeneration through paracrine signaling and growth factor release, resulting in fibroblast proliferation and tissue remodeling [2].

Wound healing is a complex cascade involving the interaction of inflammatory cells, skin fibroblasts, keratinocytes, and endothelial cells in injured tissue. These cells contribute to wound healing by releasing various chemo-cytokines, growth factors that promote cell migration to the injured area and stimulate inflammation, angiogenesis, wound contraction, and remodeling, resulting in a healthy wound-healing process. The first phase of wound healing starts with the inflammation phase, which starts within 6–8 h after injury. During this phase, platelets migrate to the tissue and release chemo-attractive cytokines; next, macrophages arrive and phagocyte/debride the tissue/organisms and set the stage for the proliferative phase. The proliferative phase starts around 5–7 days after injury and is initiated by cytokines released from macrophages (PDGF, TGF-α/β, FGF, etc.). In this stage, the formation of granulation tissue occurs with fibroblast proliferation and extracellular matrix deposition. During this phase, angiogenesis occurs, which allows leukocyte migration and provides nutrients and oxygen to develop granulation tissue. The final stage is tissue remodeling, in which wound contraction and extracellular matrix reorganization occurs over several months to years, transitioning into mature scar formation. Overall, an efficient wound-healing process results from a sufficient supply of growth factors, nutrients, cell–cell interactions, and adequate oxygenation to the tissue. Disruptions in these mechanisms, caused by conditions such as infection, malnutrition, chronic disease, or diabetes, can lead to delayed wound healing and chronic wound formation. Despite addressing systemic factors (controlling blood glucose levels, optimizing oxygenation to the tissue, providing local wound care), chronic wound care only achieves moderate success and treatment options are limited [12].

This systematic review aims to explore the use of stem cells in the treatment of chronic wounds as an ancillary method and investigate the current evidence regarding the clinical utilization of stem cells and stem cell products from various resources.

## 2. Materials and Methods

This review follows the PRISMA-S 2021 reporting standards for systematic reviews. KT and BF developed the search strategy. KT executed the searches (Figure 1) [13].

### 2.1. Information Sources

A comprehensive search of the literature for primary sources was undertaken in the following sources: Medline and PubMed Central (PubMed), Embase (Elsevier), Web of Science (Clarivate), Cochrane Library, and Google Scholar.

### 2.2. Search Strategy

Each source was searched independently. The search strategy was developed using keywords and subject headings, where available, related to the concepts of stem cells AND wound healing AND humans. The complete search strings for each database used are included in Appendix A.

No limits were applied to the searches. The search filter, NOT (“Animals” [Mesh]) NOT (“Animals” [Mesh] AND “Humans” [Mesh]), or an equivalent specific to each database, was applied to the searches to reduce the number of animal studies in the results.

The search strategy for each source was peer-reviewed by two additional academic medical librarians.

A final search was run, and references were exported on 18 December 2023. Appendix A includes the exact date and number of results for each source searched.

### 2.3. Eligibility Criteria

We established specific criteria for study inclusion in this systematic review. Studies of any design that reported on stem cell therapy and chronic wounds were considered eligible. Clinical trials, including randomized clinical trials, involving human subjects were included. Animal studies or studies that did not focus on human patients were excluded.

References were uploaded into Covidence for deduplication and screening. Two independent reviewers (BF and KR) screened the abstracts of the identified studies and reviewed the full texts of the studies that were deemed potentially eligible. Disagreements between the reviewers were resolved through consensus discussions.

### 2.4. Data Collection and Extraction Process

Data related to the application of stem cell therapy in the context of chronic wound healing were systematically extracted from each included study. The following information was recorded, if reported: study type, year of publication, treatment groups, number of participants, and the results of the study. All extracted data were documented in a Microsoft Excel spreadsheet for further analysis (Appendix A). A meta-analysis was not conducted due to the substantial heterogeneity observed among the included studies.

## 3. Results

Figure 1 shows the search results as a flow diagram. It shows the initial 22,454 results retrieved from five sources, which were narrowed via deduplication and automation to 3186 articles for title and abstract screening. Upon applying the exclusion and inclusion criteria, 3012 articles were excluded as irrelevant, leaving 170 articles for full-text screening. After reviewing the full-text articles, 126 articles were excluded, leaving 44 included studies. The excluded articles included articles on burn scars; conference abstracts; duplicate papers; case reports; animal studies; studies with the wrong setting (such as stem cell application in scar revision, chronic radiation dermatitis, tissue augmentation), wrong intervention (such as platelet-rich plasma application to chronic wounds or stem cell application to prevent critical limb ischemia), wrong outcomes (outcome scar revisions), wrong patient population (patients with scars, patients who underwent subcision, patients with alopecia), or wrong route of administration (application of fat tissue without any intervention); studies on epidermolysis bullosa; and papers for which we were not able to locate the full-text articles.

### 3.1. Skin-Derived Stem Cells

#### 3.1.1. Autologous Keratinocytes

In an initial study which demonstrated the effectiveness of keratinocytes and fibroblasts in wound healing, twelve diabetic ulcers were treated with keratinocyte epithelium and fibroblast–gelatin sponge weekly and all but one ulcer completely healed [14]. This led to a multicenter study spanning 24 U.S. centers, wherein 208 patients were randomly assigned to receive Graftskin (human fibroblasts and human epithelium) every 4 weeks (112 patients) or saline-moistened gauze (control group, 96 patients). At the 12-week follow-up, 56% of the Graftskin-treated patients achieved complete wound healing compared to 38% in the control group (*p* = 0.0042). The median time to complete closure was significantly lower for the Graftskin group (65 days) than for the control group (90 days) (*p* = 0.0026). Adverse reactions were similar, except for fewer cases of osteomyelitis and lower-limb amputations in the Graftskin group [15]. Similarly, in a randomized, controlled, multicenter study involving 314 patients, Dermagraft (human fibroblast-derived dermal substitute) was compared to a control. By week 12, 30% of the Dermagraft-treated patients achieved complete wound closure compared to 18.3% in the control group (*p* = 0.023) [16].

A study with 40 participants having grade II and III ulcers showed that the treatment group, receiving allogenic keratinocytes on polyethylene and silica microcarriers every 3 days, had a remarkable 92% reduction in wound area after 30 days, compared to 32% in the control group (*p* < 0.001). The treatment group also required fewer dressings for complete healing (9.2 + 3.2) compared to the control group (16.5 + 2.3) (*p* < 0.001) [17].

In randomized, single-blinded study by Moustafa et al. (2007), autologous keratinocytes on transfer discs were compared to a cell-free placebo for healing diabetic neuropathic ulcers. Although 18 of the 21 initially resistant ulcers responded positively to autologous keratinocyte applications, the response at 6 weeks did not achieve statistical significance [18].

In a larger multicenter trial by Ukat et al. (2007), a commercially available combination of autologous keratinocytes with fibrin sealant was compared to standard care for healing recalcitrant venous leg ulcers. After 3 months, 38.3% of patients who received autologous keratinocyte treatment achieved complete healing, in contrast to 22.4% of patients who received standard treatment (chi-square test: *p* = 0.0106) [19].

Han et al. (2009) demonstrated that primary fresh dermal fibroblasts embedded in fibrin resulted in complete healing in 83.8% of cases at 8 weeks [20]. Similarly, You et al. (2012), found that a primary foreskin keratinocyte sheet on Vaseline gauze resulted in complete healing in 100% of treated wounds [21]. Marchesi et al. (2014) reported a 70% reduction in wound area at the end of 70 days in 11 patients using primary keratinocytes in a hyaluronic acid scaffold [22]. Hwang et al. (2019) treated ulcers in 71 patients with primary foreskin keratinocyte for 12 weeks, resulting in 64.7% of patients experiencing complete wound healing within an average of 6.1 weeks [23].

#### 3.1.2. Hair Follicle-Derived Stem Cells

The hair follicle outer root sheath (ORS) is a putative source of stem cells with regeneration capacity. The ORS contains several multipotent stem cells, especially on the distal compartment of the bulge region. These stem cells can give rise to neuroectodermal and mesenchymal stem cell populations, which makes them an easily accessible source for the stem cell niche [24]. Renner et al. reported using tissue-engineered autologous epidermal sheets derived from ORS cells of patient’s hair in chronic wound healing. In their study, the sheets were placed in the wound bed and they found that complete wound closure was significantly higher in the intervention group, especially with patients that had a small ulcer area (<25 cm^2^) [25].

### 3.2. Peripheral Blood-Derived Cells/Endothelial Progenitor Stem Cells

EPCs are endothelial precursor cells involved in revascularization of injured tissue and tissue repair and have been extensively studied in ischemic heart diseases, stroke, and peripheral arterial disease [26]. Suh et al. reported that intradermal injection of EPCs stimulates the production of various cytokines that accelerate the wound-healing process in a murine model of a dermal excisional wound. When the wounds were analyzed immunohistochemically, it was shown that the EPC-stimulated wounds exhibited significantly increased monocytes/macrophages at the fifth day of injury and promoted neovascularization [27]. Di Santo et al. showed that in vitro, EPC-conditioned medium (EPC-CM) significantly inhibited the apoptosis of mature endothelial cells and promoted angiogenesis after 72 h of hypoxia in a rat aortic ring assay. In the same study, the authors demonstrated that the regenerative potential of EPC-CM was equivalent to EPC transplantation in hindlimb ischemia by inducing a substantial increase in blood flow, thus enhancing neovascularization, vascular maturation, and muscle function in rat models [28].

In 2020, a randomized double-blind clinical trial was conducted to assess the efficacy of using umbilical cord blood-derived platelet gel for treating diabetic foot ulcers. The study included 244 patients randomly assigned to intervention (PRP gel), placebo (platelet-poor plasma gel), or control (lubricant gel) groups. No significant differences were observed among the three groups in terms of wound recovery and tissue regeneration. The platelet gel group exhibited similar efficacy to both the placebo (platelet-poor plasma) and the control group. The authors concluded that while growth factors in platelet granules may aid in the wound-healing process, achieving a statistically better clinical outcome may require consideration of other factors. They also suggested that diabetic patients might not respond sufficiently, potentially due to underlying pathologic or genetic conditions [29].

In 2022, Tanaka et al. studied nine patients treated with MNC-QQ therapy. At the end of 12 weeks, six out of the ten treated ulcers achieved complete wound closure with an average closure rate of 73.2% + 40.1%. Although there was no control group for comparison, the authors concluded that the treatment increased vascular perfusion, skin perfusion pressure, and decreased pain intensity in all patients. Notably, several adverse events were observed, including cellulitis at an injection site (*N* = 1), restenosis (*N* = 4), chronic subdural hematoma (*N* =1), bedsores (*N* = 1), heterotopic ulcer (*N* = 1), urinary tract infection (*N* = 1), arthralgia (*N* = 1), dyspnea (*N* = 1), hypoperfusion (*N* = 1), labial herpes simplex (*N* = 1), diarrhea (*N* = 1), patellofemoral joint pain (*N* = 1), fever due to respiratory infection (*N* = 1), ALP elevation (hepatic–cystic system failure) (*N* = 1), and CRP elevation (*N* = 1) [30].

In 2023, Johnson et al. tested the use of platelet extracellular vesicles (pEVs) from activated platelets for treating healing biopsy sites. The investigators created two biopsy wounds on the upper arms of 11 individuals. One arm received a single subcutaneous injection of clinical pEVs (100 μg in 340 μL), while the other arm underwent standard wound care. Notably, there was no discernible distinction in the overall duration for wound closure between the treated and untreated arms. Both groups exhibited a mean healing period of 22.8 ± 8.7 days, and all wounds had fully healed within a 30-day timeframe [31].

### 3.3. Bone Marrow-Derived Mesenchymal Stem Cells (BM-MSCs)

Bone marrow-derived stem cells, encompassing both hematopoietic stem cells (HSCs) responsible for blood cell formation and mesenchymal stem cells (MSCs) with the capacity to differentiate into various cell types, including bone, cartilage, and fat cells, originate from the spongy tissue within bone cavities. While MSCs are not exclusive to bone marrow and are also found in tissues like adipose tissue, umbilical cord tissue, and synovial fluid, the bone marrow, particularly within long bones like the femur and tibia, serves as a valuable reservoir for these stem cells, making them instrumental for tissue repair across the body. Extraction of autologous bone marrow-derived cells involves a minimally invasive procedure known as bone marrow aspiration, with a specialized needle accessing the posterior iliac crest under local anesthesia. This aspirate, comprising a mix of hematopoietic and mesenchymal stem cells, holds promise for regenerative medicine. The unique regenerative properties of bone marrow-derived stem cells are crucial to optimizing their selection for the development of efficient wound-healing therapies.

Distinctive characteristics set bone marrow-derived stem cells apart from those derived from other tissues, especially in the realm of wound-healing therapy. Bone marrow serves as a rich source of diverse stem cells, including hematopoietic stem cells (HSCs) and mesenchymal stem cells (MSCs), each playing a unique role in blood cell formation and tissue regeneration. The advantageous attributes of bone marrow-derived stem cells lie in their innate ability to modulate the immune response, stimulate angiogenesis, and promote tissue repair, making them particularly well-suited for wound-healing applications. While stem cells can be sourced from alternative sites such as adipose tissue or the umbilical cord, the unique regenerative properties and versatile differentiation potential of bone marrow-derived stem cells position them as pivotal players in the development of effective therapeutic strategies for wound healing. Understanding these distinctions is essential for optimizing the selection of stem cell sources in the pursuit of targeted and efficient wound-healing therapies.

Bone marrow-derived mesenchymal stem cells (BM-MSCs) have emerged as a promising candidate for enhancing wound healing. These cells possess the ability to differentiate into multiple lineages, including cartilage, muscle, connective tissue, and adipose cells. Recent studies have demonstrated their capability to differentiate into various skin cell types, contributing significantly to wound repair. Additionally, research has shown that the application of BM-MSCs, whether through injection or topical occlusive dressing, accelerates wound healing by releasing proangiogenic factors and differentiating them into critical cell types. The therapeutic efficacy of BM-MSCs has been confirmed in human patients with chronic leg ulcers, demonstrating reduced wound size, increased vascularity, and dermal thickness. Notably, BM-MSC application has shown significant reductions in wound area as early as 2 weeks after application in patients with chronic lower-extremity wounds. These findings highlight the potential of BM-MSCs as a valuable and effective approach to advancing wound-healing therapies [8,9,12,32].

In a case–control study involving 75 patients with chronic wounds, 50 were treated with autologous bone marrow (BM) aspirate, either fresh or cultured, while 25 received daily saline dressings. Notably, both the fresh and cultured BM aspirate, even without specific identification, isolation, and selective application of stem cells, led to a significant reduction in wound surface area compared to the control group at day 7 and week 4 [11]. Another study by Bonora et al. focused on patients with diabetes and ischemic wounds. They were randomized to receive a single subcutaneous injection of plerixafor or saline in addition to standard medical and surgical therapy. The trial was terminated after an interim analysis of 50% of the target population revealed a significantly lower healing rate in the plerixafor group compared to the placebo group. In the final analysis, the plerixafor group exhibited a healing rate of 38.5%, while the placebo group showed a higher rate of 69.2% [33].

Badiavas et al. aimed to establish proof of principle that bone marrow-derived cells applied to chronic wounds could lead to wound closure. Autologous bone marrow cells were applied to chronic wounds in three patients with wounds lasting over a year. The results demonstrated complete wound closure, dermal rebuilding, reduced scarring, and successful engraftment [34].

Vojtassak et al. tested a new technique for treating chronic non-healing wounds, specifically diabetic ulcers. They utilized an autologous biograft composed of autologous skin fibroblasts on a biodegradable collagen membrane (Coladerm) combined with autologous mesenchymal stem cells (MSCs) derived from the patient’s bone marrow. The combined treatment resulted in a steady overall decrease in wound size, increased vascularity of the dermis, and an augmented dermal thickness of the wound bed after 29 days [35].

Rogers et al. presented results from a case series study involving three cases where bone marrow aspirate containing marrow-derived cells was applied or injected locally into complex lower-extremity chronic wounds of varying etiologies. Their findings suggest that topically applied and injected bone marrow aspirate may be a useful adjunct to wound simplification and ultimate closure, as evidenced by reductions in chronic wound size [36].

In a prospective randomized clinical study, Jain et al. compared the healing rates of chronic lower-limb wounds in diabetic patients treated with topically applied and locally injected bone marrow-derived cells versus whole blood (control). The study demonstrated a significant decrease in wound area by 17.4% compared to 4.84% in the control group at 2 weeks, with an average decrease of 36.4% versus 27.24% [37].

Andersen et al. conducted a study aimed at examining the safety of using a mesenchymal stem cell product for the treatment of diabetic foot ulcers. Their study involved participants who received a singular application of an allogeneic cellular product topically, which comprised mesenchymal stem cells enriched with CD362 and suspended in collagen solution. Although two individuals experienced increased exudation, it was resolved within a week of application with no further complications noted [38].

Additionally, Falanga et al. reported a successful clinical trial involving the topical application of autologous mesenchymal stem cells (MSCs) to accelerate the healing of both human and experimental murine wounds. The cultured autologous MSCs were applied using a fibrin polymer spray system, showing a strong correlation between the number of cells applied and the subsequent decrease in chronic wound size. This approach also stimulated the closure of full-thickness wounds in diabetic mice. In this particular study, there were four human subjects who underwent Mohs micrographic surgery (MMS), and their wounds were not suitable for primary closure. These patients underwent bone marrow aspiration 2 weeks prior to the procedure to allow proper in vitro establishment of the MSC cultures by the day of surgery. These cells were applied immediately after removal of the skin cancer with MMS, as well as later within 12 weeks of wound closure, which led to complete and persistent wound closure wound closure [39].

Another indication of the use of BM-MSCs has been reported as chronic graft versus host disease (GvHD). Zhou et al. reported a regression of symptoms (pain, ulceration of the skin, pliability) after intravenous application of autologous MSCs to chronic GvHD patients with leukemia, without disease recurrence [40]. Supporting the latter study, Boberg et al. reported the successful use of intravenous allogeneic MSC infusion for treating refractory chronic GvHD in 11 patients with durable responses over 12 months [41].

#### Bone Marrow-Derived Mononuclear Stem Cells

Bone marrow-derived mononuclear stem cells (BMMCs) are a heterogenous group of cells that include mature B and T cells, monocytes, endothelial progenitor cells and embryonic-like cells, and cells positive for CD 133, CD 117, and CD 34. Due to their abundance in both peripheral blood and bone marrow, BMMCs do not need in vitro expansion, and therefore are a feasible source of stem cells [42]. They have been clinically applied for chronic ulcers and their most notable property is their ability to secrete angiogenic factors. Jain et al. reported significant decrease in wound area in diabetic ulcers after application of BMMCs via spraying the wound bed after debridement compared to a control group [34]. Yamaguchi et al. reported rapid healing of intractable diabetic foot ulcers with exposed bones after treatment and grafting with epidermal sheets. In this interesting study, the authors debrided the wound with a scalpel, followed by partial excision with a bone scraper to expose the BMMCs. Later, the area was covered with an epidermal skin graft harvested from suction blisters. This combination led to the diabetic ulcers healing without the occurrence of osteomyelitis or necessity of amputation [43].

In a pilot study by Wettstein et al., autologous hematopoietic CD34+ selected stem cell suspension was injected into sacral III–IV stage pressure ulcers in three complete para- or tetraplegic patients and monitored for three weeks. They used 3D laser scanning to assess wound volume and circumference, and observed an about 50% reduction in volume versus 40% in the control side of the wound. In addition, this study did not find signs of malignant transformation at 2 years after cell application and re-debridement prior to flap coverage of wound: a possible concern, as the mechanism of healing skin wounds can be similar to that of malignancy [44].

Similarly, another study evaluated the use of bone marrow mononuclear cells (BM-MNCs) in patients with spinal cord injury and stage IV pressure ulcers. MNCs were isolated from autologous bone marrow aspiration and injected into the pressure ulcers after debridement and forming sutured-closed pouches. Overall, 19 out of the 22 patients treated achieved complete closure of their wound in a mean time of 21 days [45].

### 3.4. Adipose Tissue-Derived Mesenchymal Stem Cells

Adipose tissue-derived stem cells (ADSCs) are found in the stromal fraction of the adipose tissue. ADSCs were defined as CD45-negative, CD90-, CD73-, and CD105-positive cells [41]. Unlike other stem cells, ADSCs can easily be collected without any ethical problems and differentiate into different cell lines including adipogenic, osteogenic, chondrogenic, and myogenic cells. Thus, they are studied extensively as one of the leading sources in stem cell therapy for regenerative medicine [46,47].

ADSCs can easily adhere to plastic culture flasks and expand in vitro, and they have the capacity to differentiate into different cell lines. They have been reported to be effective in wound healing, acute graft versus host disease, and hematologic and immunologic disorders via their immunomodulatory properties [48]. In contrast to the intrusive methods required for harvesting BM-MSCs, adipose tissue is plentiful and can be easily obtained through liposuction, resulting in a less invasive process. Although flow cytometry is the conventional method for isolating AD-MSCs, autologous fat grafting can also be a valuable and practical alternative [49].

Impacts of ADSCs on neovascularization in ischemic tissue in animal models have been shown and these cells can release many potent angiogenic factors as well as the fact that they are capable of differentiating into endothelial cells, thus increasing tissue vascularization [15]. ADSCs secrete TGF-β, vascular endothelial growth factor (VEGF), keratinocyte growth factor (KGF), fibroblast growth factor 2 (FGF2), PDGF, HGF, fibronectin, and collagen I, which have been previously shown to stimulate wound-healing processes previously [16]. Studies have suggested that ADSCs can promote wound healing with paracrine activity with the aforementioned chemokines [17,18,19]. Skin wounds treated with ADSCs have been shown to exhibit enhanced healing rates and less scar formation. It was shown that the human epidermal keratinocyte migration rate was increased when co-cultured with ADSCs [20].

Exosomes or extracellular vesicles have been defined as “particles naturally released from the cell that are delineated by lipid bilayer that cannot replicate or do not contain a functional nucleus” [50]. They can be secreted from various cell types and act as intercellular communications, delivering bioactive cargos, such as proteins, lipids, nucleic acids, miRNAs, and growth factors. They regulate cell-to-cell communication and regulate metabolism and homeostasis. Increasing evidence shows that exosomes derived from ADSCs exhibit anti-inflammatory properties through inducing the polarization of macrophages to the M2 type through the STAT-3 pathway and reducing inflammation. Exosomes contain microRNSAs, which reinforce the acceleration of wound healing [51]. Additionally, ADSC exosomes promote scarless wound healing by preventing fibroblasts from differentiating into myofibroblasts and resulting in better cosmesis. These exosomes can be delivered to the tissue with injections, by being loaded into alginate hydrogel or loaded into wound dressings [52].

It was shown that autologous ADSCs combined with atelocollagen accelerated wound healing in a diabetic chronic wound model on mice via increasing the healing time, epithelization rate, granulation tissue, and vascular formation compared to a control group [53]. Altman et al. reported enhanced wound healing when full-thickness wounds were sutured with ADSC-seeded silk sutures in mice [54]. Furthermore, Blanton et al. reported better cosmesis and vascularization on porcine skin when ADSCs and platelet-rich plasma were applied together topically compared to only ADSCs or PRP applications in mice [55]. ADSCs appear to enhance wound healing via differentiation to other cell lines and paracrine activity.

A pilot study by Chopinaud et al. reported that fat grafting to hypertensive leg ulcers resulted in decreased wound healing time with better cosmesis when compared with a control ulcer in the same patient. The median wound closure rate was 73.2% and 93.1% at 3 months and 6 months of follow up. They also reported a significant increase in granulation tissue and reduction in pain without any adverse events [56].

Moon et al. examined the potential of allogeneic ADSC sheets for diabetic wound treatment. A total of 59 patients were randomized to either a control group or ADSC-sheet group. After 12 weeks of evaluation, they found that 82% of the ADSC sheet-treated patients achieved a complete wound closure, whereas 53% of patients in the control group achieved complete closure of their wounds. Parallel to later findings, the times to complete closure from the beginning of application were 28.5 and 63 days, respectively, in the ADSC sheet-treated group and the control group. They argued that this technique is a safe and effective method for treating diabetic ulcers [57].

Han et al. used processed lipoaspirate (adipose tissue cells incubated with collagenase and centrifuged) to promote diabetic wound healing. In their study, they used processed lipoaspirate (PLA) for 26 patients with diabetes and after 8 weeks of treatment, their cell proliferation and collagen synthesis rates were higher than those of the control group (44% vs. 28%). Additionally, the PLA-applied group achieved 100% complete wound closure, whereas only 62% of the control group achieved complete closure of their wounds [58]. Lee et al. similarly used autologous PLA from thromboangiitis obliterans (TAO) (n:12) patients to treat diabetes mellitus (DM) patients (n:3) with critical limb ischemia. In their study, they also used a control group to assess the quantity of stromal vascular fractions (SVFs) of adipose tissue in different groups. In a colony-forming unit assay, the SVFs of TAO and DM patients yielded less colonies and lesser proliferative capacity than that of healthy controls. Multiple intramuscular injections of PLA were administered with a mean follow up of 6 months, resulting in a clinical improvement in 66.7% of the patients. Only five patients required minor amputations and all amputation sites healed completely. All diabetic patients’ wounds completely healed (100% wound closure rate in this subgroup) and pain significantly decreased in all patients. Digital subtraction angiography showed a significant collateral vascular network in the critically ischemic limbs by the end of 6 months [59]. Maslowski et al. used autologous ADSCs for the treatment of chronic venous stasis ulcers. An SVF was injected subcutaneously to the periphery of the ulcers and to the wound bed in addition to traditional local treatments and the wounds were measured planimetrically for 6 months. Overall, improvement was observed in 75% of the patients and no serious side effects were observed. Eight of the patients were healed completely, nine of the patients showed a >50% reduction in the ulcer area, and five patients showed no improvement [60]. Marino et al. used autologous ADSCs to treat chronic arterial leg ulcers. They applied the SVF to the edges of the wound with a depth of 1 cm. All patients showed a reduction in the diameter and depth of their ulcers and 6 out of 10 patients experienced complete closure of their wounds [61].

Recently, an automatized processing system for isolating ADSCs has been developed. The system simply standardizes the processing of human adipose tissue to harvest and concentrate in a real-time bedside manner. The system reliably and reproducibly generates ADSCs from collected adipose tissues via liposuction and it allows the application of the product in approximately 1.5 h. This system has been used in the cosmetics field to rejuvenate the face and perform breast augmentation and reconstruction in small studies with success [62]. Moreover, this system can simply be utilized for acute radiation dermatitis at bedsides to prevent radiation-related skin side effects in cancer patients [63].

In a randomized controlled trial, 32 individuals suffering from chronic plantar ulcers due to leprosy were enrolled. Following thorough clinical examination and initial debridement, the participants were randomly assigned to two cohorts: one group (n = 16) received topical application of a conditioned medium from adipose mesenchymal stem cells, while the other group (n = 16) was treated solely with framycetin gauze dressing applied every three days for a duration of eight weeks. Throughout this period, ulcer size, as well as any adverse reactions or complications, were monitored on a weekly basis. The findings revealed a consistent increase in the healing percentage across all groups each week. Notably, statistical distinctions between the two groups (*p* < 0.05) became apparent from the second week onward for the reduction in mean ulcer size and from the third week onward for the reduction in mean ulcer depth. Importantly, no adverse reactions or complications were reported throughout the study period [64].

Furthermore, Del Papa et al. reported on ADSC fraction injection for treatment-resistant digital ulcers (DUs) in systemic sclerosis, which resulted in a reduction in pain and analgesic consumption, faster healing rate of the DUs, significant increase in the number of capillaries, and reduction in arterial resistance, leading to favorable outcomes in the intervention group and presenting stem cell treatment as a promising method [65].

A new indication for the use of SVFs of adipose tissue has been reported to be vulvar lichen sclerosis. Autologous fat grafting enriched with adipose tissue derived SVF cell injections to the vulvar skin resulted in a significant decrease in global scoring for burning, pain, dryness, and distress associated with sexuality and resulted in better cosmesis in the area [66].

### 3.5. Placental/Umbilical/Wharton’s Jelly-Derived Mesenchymal Stem Cells (WJSCs)

Human embryo fibroblast-, Wharton’s jelly-, umbilical cord-, and human placenta-derived stem cells have shown significant potential in improving healing rates and reducing wound size.

#### 3.5.1. Human Embryo Fibroblast

Sedov’s (2006) study focused on the efficacy of “Foliderm” in treating venous trophic ulcers. Patients either had post-thrombophlebitic disease or varicose disease; the treatment group was 23 patients and the control group had 25 patients [67]. The treatment group demonstrated a 100% healing rate, compared to 86% for varicose and 78% for post-thrombophlebitic ulcers in the control group. Additionally, the treatment group experienced significantly shorter average healing periods [67].

#### 3.5.2. Wharton’s Jelly-Derived Stem Cells

Between the tenth and twelfth weeks of embryogenesis, the extraembryonic mesoderm undergoes development, creating a protective extracellular structure for the umbilical cord known as Wharton’s jelly [68]. Wharton’s jelly, a gel-like substance within the umbilical cord, is abundant in proteoglycans, particularly hyaluronic acid, and chondroitin sulfate. Mesenchymal stem cells present in Wharton’s jelly express c-KIT and exhibit telomerase activity, characteristics indicative of their role as a source of stem cells [59]. The ease of extraction of these cells post delivery makes them a convenient stem cell source, accompanied by fewer ethical considerations compared to alternative stem cell resources [69].

In a 2019 study conducted in Iran (Hashemi, 2019), the healing potential of Wharton’s jelly-derived stem cells was evaluated in five patients with chronic diabetic wounds. There was no control group, but the authors concluded that acellular amniotic membrane seeded with WJ-MSCs significantly reduced wound healing time and size on days 6 and 9 [70].

#### 3.5.3. Human Umbilical Cord Mesenchymal Stem Cells

Zhang (2022) investigated the topical and intravenous administration of human umbilical cord mesenchymal stem cells (hUC-MSC) for diabetic foot ulcers (DFUs) and peripheral arterial disease (PAD). This study involved 14 patients with PAD and incurable DFUs. In this phase I pilot study, the safety and efficacy of hUC-MSC administration were assessed against conservative treatments. Topical and intravenous hUC-MSCs at 2 × 105 cells/kg (up to 1 × 107 cells per dose) were administered to all 14 patients. There was no control, but the authors report that the treatment resulted in ulcer closure and symptom alleviation, but no direct evidence of vessel obstruction improvement was observed. Rehospitalization for the DFUs lasted 2.0 + 0.6 years, and all patients survived without amputation within three years after treatment. Safety assessments recorded only two cases of transient fever [71].

Expanding upon these findings, Tan’s (2023) study centered on a 10% secretome from human umbilical cord mesenchymal stem cells (SM-hUCMSCs) in gel form. The study involved forty-one patients with chronic ulcers, including diabetic ulcers. The patients’ ulcers demonstrated significant improvements in ulcer length, width, and area after the intervention. There was no control for comparison, but the change between the beginning and end of the intervention was statistically significant (*p* value < 0.05) [72].

Furthermore, the value of hUCMSCs as a systemic infusion has shown to be a promising agent in Stevens–Johnson syndrome (SJS). According to the study by Li and colleagues, three patients with SJS with severe skin and mucosa lesions that were unresponsive to systemic prednisolone and supportive therapies exhibited interruptions to their skin blistering as well as recovery of their blood chemical abnormalities within 12 days of hUCMSC infusion. They concluded the potential capacity of blocking the apoptosis in keratinocytes of hUCMSCs. Therefore, hUCMSC infusion treatment might be a new salvage therapy for cutaneous severe drug eruptions in order to prevent further damage [73].

#### 3.5.4. Human Placenta/Human Amniotic Membrane

In 2017, Dehghani et al. conducted a randomized clinical trial to assess the efficacy of using grafting with cryopreserved amniotic membrane for the treatment of pressure ulcers in 24 patients with second- and third-stage pressure ulcers who required split-thickness skin grafts. Patients were assigned to a treatment group or a control group, which received treatment with local Dilantin powder. The amnion group demonstrated a significantly higher rate of complete healing (*p* < 0.001) [74].

In a prospective study, Farivar’s (2019) study using the cryopreserved placental tissue wound matrix achieved complete healing in 53% of refractory venous leg ulcers (VLUs) among 21 patients [75].

In 2020, Suzdaltseva conducted a trial on locally delivered umbilical cord mesenchymal stromal cells (UCMSCs) to reduce chronic inflammation in long-term non-healing wounds, involving 108 patients with chronic wounds of various etiologies. The treatment group (n = 59) received a single local subcutaneous infusion of UCMSCs around the wound periphery, while the placebo group (n = 49) received a placebo. The patients treated with the UCMSCs exhibited notable growth of their granulation tissue, improved blood microcirculation, and a reduction in wound size compared to the placebo group [76].

Expanding on these findings, Meamar et al. (2021) conducted a clinical trial of twenty-eight patients with diabetic foot ulcers (DFUs) randomly assigned to three groups over a 12-week period. The two treatment groups involved the use of human placental-derived mesenchymal stem cells (hPDMSCs) without PRP gel (Group A) and with PRP gel (Group B), while the third group received standard wound care as the control group (Group C). The wound size reduction was 66% in Group A, 71% in Group B, and 36% in the control group (Group C). Significant differences in wound closure and pain-free walking distance were observed in Groups A and B compared to the control group (Group C) (*p* < 0.05). There was no significant difference between Groups A and B [77].

Similarly, Rezaei-Nejad et al. (2023) tested the efficacy of using freeze-dried human amniotic membrane allograft gel (LAMG) in a clinical trial among 18 patients with chronic diabetic foot ulcers (DFUs) in their study. The patients were randomly assigned to two groups: the LAMG group (n = 9) and the placebo group (n = 9). After 9 weeks, the study revealed that the freeze-dried human amniotic membrane allograft treatment led to a mean wound size reduction of 73.4%, in contrast to the control group’s mean reduction of 13.1% + 10.1% [78].

### 3.6. Inducible Pluripotent Stem Cells

Inducible pluripotent stem cells (iPSCs) are pluripotent stem cells derived from somatic donor cells that are generated via overexpression of Oct4, Klf4, Sox2, and c-myc transcription factors in adult somatic cells harvested from healthy objects. iPSCs have the capacity to differentiate into and repopulate all cell types found in the skin [9]. Although there are some concerns regarding their safety and regenerative capacity, iPSCs have been investigated in clinical trials of disease modeling, including cardiomyopathy, autism spectrum disorder, coronary artery disease, and cystic fibrosis [79]. Human-induced pluripotent fibroblasts, human-induced pluripotent mesenchymal stem cells, and human-induced pluripotent stem-cell-derived vesicles have the potential to accelerate wound healing [80]. Two recent studies from Sebastiano et al. and Umegaki-Arao et al. reported the successful use of human keratinocyte-derived iPSCs to reconstitute skin in vitro for recessive dystrophic epidermolysis bullosa [81,82]. Clayton et al. showed that injection of iPSC-derived endothelial cells promoted angiogenesis and accelerated wound closure in a murine excisional wound model [82]. These findings can enable the generation of iPSC-based cutaneous substitutes that include epidermal appendages and may be attractive as a therapeutic option in wound healing. However, there are concerns regarding the safety of iPSCs: since iPSCs can differentiate into cells from any of the three germ layers, they carry a risk for teratoma formation. Different strategies, including using viral vectors to deliver induced pluripotent stem cell vesicles, are under investigation to overcome this risk [80].

## 4. Discussion

Additional research is needed to understand the complex molecular mechanisms through which MSCs are involved in wound healing [83]. In vivo, MSCs migrate towards injury sites when prompted by chemotactic signals that modulate inflammation. The differentiation of MSCs aids in the regeneration of damaged tissue and their paracrine signaling regulates local cellular responses to injury [84]. MSC paracrine signaling is thought to be one of the primary mechanisms underlying the beneficial effects of MSCs in healing wounds, leading to inflammation reduction, the promotion of angiogenesis, and the stimulation of cell migration and proliferation. Moreover, the migration of mesenchymal cells, as well as the interplay between pro- and anti-inflammatory cytokines, plays an important role in regulating wound repair processes [85].

Numerous chemical and natural compounds have been identified for their potential to enhance the migratory capabilities of cells while maintaining their regenerative potential and differentiation capacity [40,86]. Chemical compounds like sphingosine-1-phosphate (S1P) and stromal cell-derived factor-1 (SDF-1) have shown promise in promoting cell migration by activating specific signaling pathways involved in cell movement, such as the PI3K/Akt and MAPK pathways [87]. Additionally, natural compounds such as epigallocatechin-3-gallate (EGCG) found in green tea and curcumin derived from turmeric have demonstrated the ability to modulate cell migration through their anti-inflammatory and antioxidant properties without compromising stem cell identity [88].

To address the issue of fetal bovine serum (FBS) in stem cell transplantation, serum-free or defined media formulations supplemented with synthetic growth factors and nutrients can be used in stem cell transplantation [89]. Similarly, exploring xenogeneic-free components and human-derived alternatives will help to eliminate the use of animal-origin products [90]. Bioreactor culture systems are also being utilized for large-scale stem cell expansion in these serum-free or defined media, promoting standardized and reproducible culture conditions for therapeutic applications [89].

## 5. Conclusions

Stem cell transplantation is a potential therapeutic approach in wound healing. Transplanted stem cells not only differentiate into multiple skin cell types, but also provide cytokines and growth factors required for wound healing, resulting in increased angiogenesis. Thus, this approach may be regarded as an attractive option for intractable wounds that cause major clinical problems, especially chronic lower-leg wounds, and may decrease the overall cost burden on the system and improve patients’ quality of life.

## Figures and Tables

**Figure 1 ijms-25-03006-f001:**
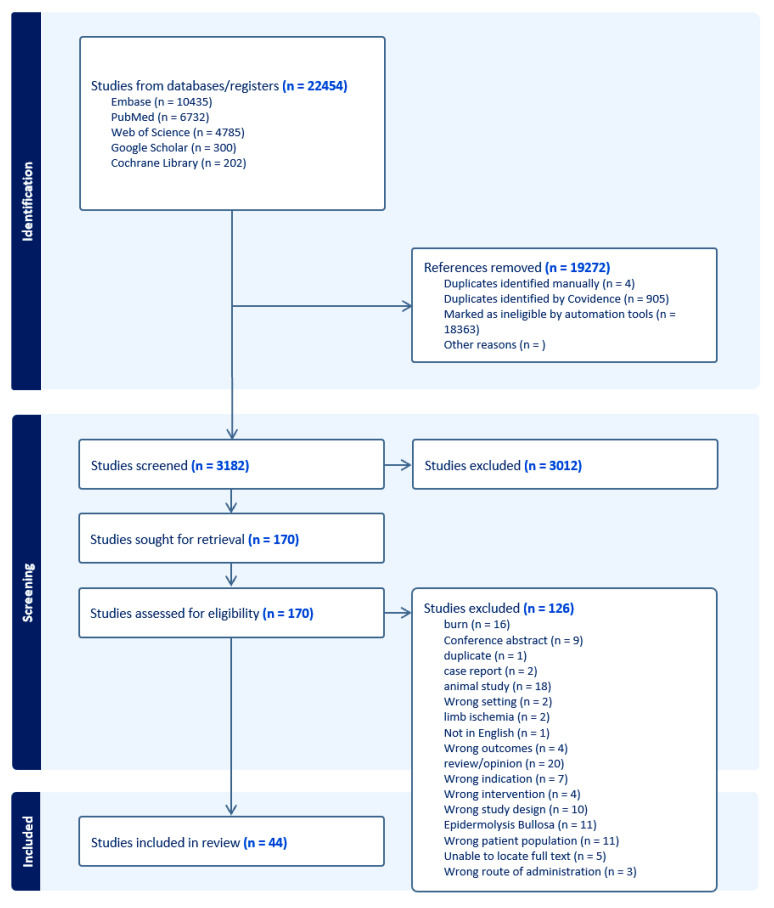
PRISMA Flow Diagram; This diagram shows the systematic process we followed to include papers captured by our search.

## Data Availability

The original contributions presented in the study are included in the article/Appendix A, further inquiries can be directed to the corresponding author/s.

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
