# Peer review of "The Efficacy of Stem Cells in Wound Healing: A Systematic Review"

_ijms, 2024, doi:10.3390/ijms25053006_

Round 1

Reviewer 1 Report

Comments and Suggestions for Authors

Authors should revise abstract and make it more concise filled with information.

Authors provide additional statistic analysis about studies and their details.

Authors should also discuss what is the outcome of relevant study and how it can help for further studies.

Authors should provide info about what is current and future therapy options, clinical trials or extensively effective state of art approaches.

Authors should expand discussion section about how their work will provide and improve future therapies.

Author Response

Authors should revise abstract and make it more concise filled with information.

Thank you for your suggestion. We have rewritten the abstract to be more concise. 

Authors provide additional statistic analysis about studies and their details.

Thank you for your comment. The purpose of our study was to provide the reader with a systematic review of the literature. Future meta-analyses are needed to better quantify the associated risk. We have added this limitation in the limitation paragraph of our study. 

We also provided an excel sheet with all the included studies and their determinants.

Authors should also discuss what is the outcome of relevant study and how it can help for further studies.

Thank you for your careful review of our work. We have added a paragraph in the discussion that summarizes the outcome of the current study and how it can help future studies. 

Authors should provide info about what is current and future therapy options, clinical trials or extensively effective state of art approaches.

We have already discussed all human clinical trials in this current work. Right now, stem cell application in wound healing is not FDA approved. This paper encompasses all the active human clinical studies of stem cells in chronic wound healing.

Authors should expand discussion section about how their work will provide and improve future therapies.

Discussion section added.

Reviewer 2 Report

Comments and Suggestions for Authors

The review proposal is interesting and will be of great importance in new studies using stem cells in wound healing in the skin. The text is exquisite, and to maintain this aspect, recommend a review of the bibliographical references cited. For example, in line 78, an extensive text presents data from the article by Werdin et al. (2008), which deals with chronic injuries with indications of chronic wound care it´s not discussed, we recommend indicating the elegant text of Rodrigues et al. (Rodrigues et al., 2019) or Tottoli et al. (Tottoli et al., 2020) or Aitcheson et al. (Aitcheson et al., 2021); some good quality texts can help in presenting to fixed this problems in the introduction. Of course, it´s necessary to review all text.

Within the extended definition of eligible articles – In Figure 1 – there are several times the terms Wrong setting, wrong outcomes, and others. This parameter is not described in the methodology for these wrong “mechanisms” in the text.

Some caveats must be added to the conclusion since most treatments with stem cells do not present a statistical difference in the control parameters, even though they reduce the time needed to repair the skin lesion. The importance of reducing repair time must be highlighted, even though it is not statistically significant with treatment controls.

Aitcheson, S.M., Frentiu, F.D., Hurn, S.E., Edwards, K., Murray, R.Z., 2021. Skin wound healing: Normal macrophage function and macrophage dysfunction in diabetic wounds. Molecules 26. https://doi.org/10.3390/molecules26164917

Rodrigues, M., Kosaric, N., Bonham, C.A., Gurtner, G.C., 2019. Wound Healing: A Cellular Perspective. Physiol Rev 99, 665–706. https://doi.org/10.1152/physrev.00067.2017.-Wound

Tottoli, E.M., Dorati, R., Genta, I., Chiesa, E., Pisani, S., Conti, B., 2020. Skin wound healing process and new emerging technologies for skin wound care and regeneration. Pharmaceutics 12, 1–30. https://doi.org/10.3390/pharmaceutics12080735

Author Response

The review proposal is interesting and will be of great importance in new studies using stem cells in wound healing in the skin. The text is exquisite, and to maintain this aspect, recommend a review of the bibliographical references cited. For example, in line 78, an extensive text presents data from the article by Werdin et al. (2008), which deals with chronic injuries with indications of chronic wound care it´s not discussed, we recommend indicating the elegant text of Rodrigues et al. (Rodrigues et al., 2019) or Tottoli et al. (Tottoli et al., 2020) or Aitcheson et al. (Aitcheson et al., 2021); some good quality texts can help in presenting to fixed this problems in the introduction. Of course, it´s necessary to review all text.

Within the extended definition of eligible articles – In Figure 1 – there are several times the terms Wrong setting, wrong outcomes, and others. This parameter is not described in the methodology for these wrong “mechanisms” in the text.

We have included the description of the work we did when we excluded studies in the material and methods section.

Some caveats must be added to the conclusion since most treatments with stem cells do not present a statistical difference in the control parameters, even though they reduce the time needed to repair the skin lesion. The importance of reducing repair time must be highlighted, even though it is not statistically significant with treatment controls.

We have added this at the end of the paper.

Aitcheson, S.M., Frentiu, F.D., Hurn, S.E., Edwards, K., Murray, R.Z., 2021. Skin wound healing: Normal macrophage function and macrophage dysfunction in diabetic wounds. Molecules 26. https://doi.org/10.3390/molecules26164917

Rodrigues, M., Kosaric, N., Bonham, C.A., Gurtner, G.C., 2019. Wound Healing: A Cellular Perspective. Physiol Rev 99, 665–706. https://doi.org/10.1152/physrev.00067.2017.-Wound

Tottoli, E.M., Dorati, R., Genta, I., Chiesa, E., Pisani, S., Conti, B., 2020. Skin wound healing process and new emerging technologies for skin wound care and regeneration. Pharmaceutics 12, 1–30. https://doi.org/10.3390/pharmaceutics12080735

Reviewer 3 Report

Comments and Suggestions for Authors

The Review "The Potential Role of Stem Cells in Wound Healing: A Systematic Review" by Banu Farabi MD, Katie Roster, Rahim Hirani, Katharine Tepper MLIS, Mehmet Fatih Atak MD, Bijan Safai, MD, DSc, FAAD, is about Stem cell transplantation as a potential therapeutic approach in wound healing process. The review is well structured and interesting because the authors evidenced that transplanted stem cells not only differentiate into multiple skin cell types, but also provide cytokine and growth factors required for wound healing resulting increased angiogenesis.

The review could be enhanced by implementing the following suggestions:

1. In the various systems analyzed, the authors deal marginally with secretome analysis. Please deepen and possibly summarize the effects in dedicated tables.

2. The authors could add a paragraph that relates mesenchymal cell migration to chronic inflammatory diseases. For example, recently several articles have highlighted the existence of a population of mesenchymal cells in nasal polyps whose migratory capacities are important in relapsing processes. The authors could therefore investigate the impact of pro- and anti-inflammatory cytokines on wound repair processes.

3. The authors could devote a paragraph to chemical and natural compounds capable of stimulating the migratory capacities of cells without affecting their stemness characteristics.

4. In addition, for stem cell transplantation, how could the problem of growth in FBS be solved, considering its animal origin?

5. The authors could complete their manuscript by adding a graphical abstract.

6. The Potential Role of Stem Cells in Wound Healing: A Systematic Review "Efficacy of Stem Cells in Wound Healing: A Systematic Review"

Comments on the Quality of English Language

Minor editing of English language required

Author Response

The Review "The Potential Role of Stem Cells in Wound Healing: A Systematic Review" by Banu Farabi MD, Katie Roster, Rahim Hirani, Katharine Tepper MLIS, Mehmet Fatih Atak MD, Bijan Safai, MD, DSc, FAAD, is about Stem cell transplantation as a potential therapeutic approach in wound healing process. The review is well structured and interesting because the authors evidenced that transplanted stem cells not only differentiate into multiple skin cell types, but also provide cytokine and growth factors required for wound healing resulting increased angiogenesis.

The review could be enhanced by implementing the following suggestions:

  1. In the various systems analyzed, the authors deal marginally with secretome analysis. Please deepen and possibly summarize the effects in dedicated tables.

Secrotomes are mentioned briefly since this is an emerging field, we were not able to find specific studies investigating their efficacy in human skin in chronic wound healing except one study by Tan et al., there were a couple of clinical studies regarding alopecia, since this was out of the scope of this review, secrotomes are not discussed in detail.

  1. The authors could add a paragraph that relates mesenchymal cell migration to chronic inflammatory diseases. For example, recently several articles have highlighted the existence of a population of mesenchymal cells in nasal polyps whose migratory capacities are important in relapsing processes. The authors could therefore investigate the impact of pro- and anti-inflammatory cytokines on wound repair processes.

Thank you for your thoughtful review and  suggestion. In response to your comment, we have included a new paragraph in the manuscript discussing the relevance of paracrine signlaing of mesenchymal cells which leads to  migration, as well as the promotion of pro and anti-inflammatory cytokines. Thank you again for your valuable feedback.

Additional research is needed to understand the complex molecular mechanisms through which MSCs are involved in wound healing (Lau, 2009).  In vivo, MSCs migrate towards injury sites when prompted by chemotactic signals that modulate inflammation. The differentiation of MSCs aids in the regeneration of damaged tissue and their paracrine signaling regulates local cellular responses to injury (Newman, 2009). MSC paracrine signaling is thought to be one of the primary mechanism underlying the beneficial effects of MSCs in wound healing wounds, leading to inflammation reduction, promotion of angiogenesis, and stimulation of cell migration and proliferation. Moreover, the migration of mesenchymal cells, as well as the interplay between pro and ant iinflammatory cytokines plays an important role in regulating wound repair processes (Gnecchi, 2008). 

Lau K, Paus R, Tiede S, Day P, Bayat A. Exploring the role of stem cells in cutaneous wound healing. Exp Dermatol. 2009 Nov;18(11):921-33. doi: 10.1111/j.1600-0625.2009.00942.x. Epub 2009 Aug 27. PMID: 19719838.

Newman RE, Yoo D, LeRoux MA, Danilkovitch-Miagkova A. Treatment of inflammatory diseases with mesenchymal stem cells. Inflamm Allergy Drug Targets. 2009 Jun;8(2):110-23. doi: 10.2174/187152809788462635. PMID: 19530993.

Gnecchi M, Zhang Z, Ni A, Dzau VJ. Paracrine mechanisms in adult stem cell signaling and therapy. Circ Res. 2008 Nov 21;103(11):1204-19. doi: 10.1161/CIRCRESAHA.108.176826. PMID: 19028920; PMCID: PMC2667788.

  1. The authors could devote a paragraph to chemical and natural compounds capable of stimulating the migratory capacities of cells without affecting their stemness characteristics.

Thank you for taking the time to review our work. We have added the below paragraph to our discussion section of our text. 

“Numerous chemical and natural compounds have been identified for their potential to enhance the migratory capabilities of cells while  maintaining their regenerative potential and differentiation capacity.  (Zhou, 2021). Chemical compounds like sphingosine-1-phosphate (S1P) and stromal cell-derived factor-1 (SDF-1) have shown promise in promoting cell migration by activating specific signaling pathways involved in cell movement, such as the PI3K/Akt and MAPK pathways (Smith, 2017). Additionally, natural compounds such as epigallocatechin-3-gallate (EGCG) found in green tea and curcumin derived from turmeric have demonstrated the ability to modulate cell migration through their anti-inflammatory and antioxidant properties without compromising stem cell identity (Almaroodi, 2020).”

Zhou T, Yuan Z, Weng J, Pei D, Du X, He C, Lai P. Challenges and advances in clinical applications of mesenchymal stromal cells. J Hematol Oncol. 2021 Feb 12;14(1):24. doi: 10.1186/s13045-021-01037-x. PMID: 33579329; PMCID: PMC7880217.

Smith P, O'Sullivan C, Gergely P. Sphingosine 1-Phosphate Signaling and Its Pharmacological Modulation in Allogeneic Hematopoietic Stem Cell Transplantation. Int J Mol Sci. 2017 Sep 21;18(10):2027. doi: 10.3390/ijms18102027. PMID: 28934113; PMCID: PMC5666709.

Almatroodi SA, Almatroudi A, Khan AA, Alhumaydhi FA, Alsahli MA, Rahmani AH. Potential Therapeutic Targets of Epigallocatechin Gallate (EGCG), the Most Abundant Catechin in Green Tea, and Its Role in the Therapy of Various Types of Cancer. Molecules. 2020 Jul 9;25(14):3146. doi: 10.3390/molecules25143146. PMID: 32660101; PMCID: PMC7397003.

  1. In addition, for stem cell transplantation, how could the problem of growth in FBS be solved, considering its animal origin?

Thank you for your comment and for raising this important issue. We have added a paragraph to the text discussing alternatives to fetal bovine serum. 

“To address the issue of fetal bovine serum (FBS) in stem cell transplantation, serum-free or defined media formulations supplemented with synthetic growth factors and nutrients can be used in stem cell transplantation (Duarte, 2023). Similarly, exploring xenogeneic-free components and human-derived alternatives will help eliminate animal-origin products (Diez, 2015). Bioreactor culture systems are also being utilized for large-scale stem cell expansion in these serum-free or defined media, promoting standardized and reproducible culture conditions for therapeutic applications (Duarte, 2023).”

Díez JM, Bauman E, Gajardo R, Jorquera JI. Culture of human mesenchymal stem cells using a candidate pharmaceutical grade xeno-free cell culture supplement derived from industrial human plasma pools. Stem Cell Res Ther. 2015 Mar 13;6(1):28. doi: 10.1186/s13287-015-0016-2. PMID: 25889980; PMCID: PMC4396121.

Duarte AC, Costa EC, Filipe HAL, Saraiva SM, Jacinto T, Miguel SP, Ribeiro MP, Coutinho P. Animal-derived products in science and current alternatives. Biomater Adv. 2023 Aug;151:213428. doi: 10.1016/j.bioadv.2023.213428. Epub 2023 Apr 24. PMID: 37146527.

  1. The authors could complete their manuscript by adding a graphical abstract.
  2. The Potential Role of Stem Cells in Wound Healing: A Systematic Review "Efficacy of Stem Cells in Wound Healing: A Systematic Review"

Thank you for your suggestions. We have changed the title to "Efficacy of Stem Cells in Wound Healing: A Systematic Review"

Round 2

Reviewer 1 Report

Comments and Suggestions for Authors

Authors improved manuscript with additions. I that format it is suitable for publication.

Reviewer 3 Report

Comments and Suggestions for Authors

The authors have significantly improved the manuscript by introducing new paragraphs and new food for thought. The manuscript is now to be considered publishable on IJMS.

Comments on the Quality of English Language

Minor editing of English language required